# Assessment of immunogenicity and protection induced by COBRA HA vaccines formulated with Infectimune in young and elderly ferrets

Xiaojian Zhang[1,2], Matthew H. Thomas[3], Amanda Lynch[3], Hua Shi[1,2], Ted M. Ross[1,2,3,4]*

**1** Center for Vaccines and Immunology, University of Georgia, Athens, Georgia, United States of America, **2** Department of Infectious Diseases, University of Georgia, Athens, Georgia, United States of America, **3** Florida Research and Innovation Center, Cleveland Clinic, Port Saint Lucie, Florida, United States of America, **4** Department Microbial Sciences in Health, Lehner Research Institute, Cleveland Clinic, Cleveland, Ohio, United States of America

* rosst7@ccf.org

## Abstract

Influenza viruses cause seasonal epidemics resulting in substantial morbidity and mortality, particularly in the elderly population. Although annual high-dose or adjuvanted influenza vaccines help maintain antibody titers in individuals aged 65 years and older, vaccine-mismatched strains, especially H3N2 influenza viruses, often lead to severe disease. Therefore, a universal influenza vaccine capable of eliciting broadly protective immunity in older individuals is urgently needed. Here, Computationally Optimized Broadly Reactive Antigen (COBRA) based H1 and H3 hemagglutinin (HA) vaccines formulated with the clinical-stage investigational immune modulator Infectimune were evaluated in pre-immune young (9 months) and elderly (50–71 months) ferrets to assess age-related differences in antibody responses. COBRA HA vaccination elicited robust H1- and H3- HA-specific antibody responses in both age groups. Following a single immunization, young ferrets developed significantly higher IgG titers than elderly ferrets, however, booster immunization with COBRA HA vaccines formulated with Infectimune increased anti-HA antibody titers in elderly animals to levels comparable to those of young ferrets. Upon influenza virus challenge, pre-immune young ferrets were largely protected against virally induced disease with minimal weight loss and reduced viral shedding, regardless if the ferrets were vaccinated with COBRA HA proteins or mock vaccinated. In contrast, only elderly ferrets receiving COBRA HA vaccines formulated with Infectimune were protected following challenge with significantly less body weight loss and lower nasal viral titers. These findings demonstrate that COBRA HA vaccines induce broad antibody responses in young and aged hosts and the addition of Infectimune enhanced vaccine efficacy in aged ferrets.

Data availability statement: Data is available from https://www.immport.org/shared/study/SDY3368/summary The accession number is: ImmPort Study Accession: SDY3368.

Funding: This work was supported by the Collaborative Influenza Vaccine Innovations Centers (CIVICs) program of the National Institute of Allergy and Infectious Diseases, National Institutes of Health, Department of Health and Human Services, under contract 75N93019C00052. T.M.R. is also supported in part as a Georgia Eminent Scholar by the Georgia Research Alliance (GRA-001). The funders had no role in study design, data collection and analysis, decision to publish, or preparation of the manuscript.

Competing interests: The authors have declared that no competing interests exist.

## Introduction

Influenza viruses circulate among both animals and humans, causing seasonal epidemics and occasional pandemics that pose a major challenge to global public health [1,2]. These enveloped viruses possess a segmented, negative-sense RNA genome encoding at least 11 proteins, among which hemagglutinin (HA) and neuraminidase (NA) are the major surface glycoproteins responsible for eliciting protective immune responses [3]. Consequently, HA and NA are the primary targets for influenza vaccine development [4].

Vaccination remains the most effective strategy for controlling influenza virus infection. Currently available influenza vaccines include split inactivated vaccines (produced in eggs or cell culture) for individuals across a wide range of ages and live-attenuated influenza vaccines, which are approved for use in healthy individuals aged 2–49 years. High-dose and adjuvanted influenza vaccines are recommended for adults aged 65 years and older, while recombinant influenza vaccines are approved for individuals aged 18 years and older [5–10]. However, despite widespread immunization, protection against influenza in the elderly remains suboptimal due to the rapid antigenic evolution of influenza viruses and age-related declines in immune function [3,11,12]. Several strategies are in development for a universal influenza vaccine capable of eliciting broadly protective antibodies to overcome the antigenic diversity of circulating strains with the goal of providing broader and longer-lasting immunity [13–16]. However, most of these next-generation vaccines have been evaluated primarily in young adult animal models and their effectiveness in populations with impaired or aged immune systems, such as the elderly, remains largely unknown.

A deeper understanding of the mechanisms underlying reduced vaccine responsiveness in the elderly, across tissue, cellular, and molecular levels, is critical for the development of more effective vaccines for this population. As individuals age, they have diminished immune responses to novel pathogens and vaccine antigens due to the progressive decline in both innate and adaptive immune function [12]. Aging is associated with thymic involution and reduced bone marrow function, resulting in decreased thymic output of mature T cells, impaired B cell maturation, and increased bone marrow adiposity that further compromises hematopoiesis [11,12,17]. In addition to these changes in primary immune organs, secondary lymphoid tissues, such as the spleen and lymph nodes, also undergo structural and functional deterioration. For example, the number and size of germinal center follicles decline with age, accompanied by reduced CXCL13 (B cell–attracting chemokine-1, BCA-1) expression and impaired B cell migration into follicles [12]. At the cellular level, aging shifts the immune milieu toward a pro-inflammatory state, characterized by reduced antibody avidity in B cells and a skewed repertoire of CD4+ and CD8+T cell populations [11,17]. Under this inflammatory environment, elderly individuals often exhibit excessive pulmonary infiltration of pro-inflammatory cells, contributing to cytokine storm-like responses following viral infection and increased susceptibility to secondary bacterial pneumonia [12,18–21]. Collectively, these age-associated immunological changes impair the ability to mount effective vaccine responses and render older

adults more vulnerable to severe influenza disease [11,12]. Therefore, evaluating next-generation vaccine platforms, such as COBRA-based hemagglutinin (HA) vaccines, in aged animal models is essential to determine their potential to overcome immune senescence and enhance protective immunity in elderly populations.

To address these age-associated limitations in immune responsiveness, COBRA HA recombinant vaccines were evaluated in young (9 months) and elderly (50–71 months) ferrets with pre-existing immunity to historical influenza viruses. This model allowed assessment of how age and prior influenza exposure influenced vaccine-induced antibody breadth and potency. Following vaccination, all ferrets were challenged with an A(H1N1) influenza virus to determine the degree of protection conferred by the vaccines.

## Materials and methods

### Viruses and cells

The seeds of influenza viruses were from either the Influenza Reagents Resource (IRR), BEI Resources, or the Centers for Disease Control (CDC). The stock viruses were propagated in embryonated 10-day-old specific pathogen-free (SPF) chicken eggs as previously described [22]. Allantoic fluid containing virus was aliquoted into 1 mL/vial and stored at −80°C until use. Prior to vaccination, animals were infected with two historical influenza viruses A/Singapore/06/1986 (SG/86, H1N1) and A/Panama/2007/1999 (PN/99, H3N2) to elicit an anti-influenza pre-immune state. Vaccinated animals were challenged with influenza virus A/Brisbane/02/2018 (BR/18, H1N1) to evaluate the protection of vaccination. To evaluate the antibody responses elicited by vaccination, the following viruses were used in the hemagglutination inhibition (HAI) assays. The H1N1 virus panel includes: BR/18, A/Guangdong/SWL1536/2019 (GD/19), A/California/07/2009 (CA/09). The H3N2 virus panel includes: A/Tasmania/503/2020 (TS/20), A/Hong Kong/2671/2019 (HK/19), A/South Australia/34/2019 (SA/19), A/Kansas/14/2017 (KS/17), A/Singapore-IFNIMH-16-0019/2016 (SG/16), A/Hong Kong/4801/2014 (HK/14), A/Switzerland/9715293/2013 (SW/13).

Madin–Darby canine kidney (MDCK) cells (Sigma, St. Louis, MO, USA) were used in influenza viral plaque assay as previously described [23]. The growth medium for MDCK cells is Dulbecco's modified Eagle's medium (DMEM; Thermo Fisher Scientific, Waltham, MA, USA) with 10% heat-inactivated fetal bovine serum (FBS; Bio-Techne, Flowery Branch, GA, USA) and 1% penicillin-streptomycin (P/S; Thermo Fisher Scientific, Waltham, MA, USA).

### Vaccines and proteins

Recombinant soluble proteins used for vaccination include: H1 COBRA HA (Y2) and H3 COBRA HA (NG2, NG5, NG7, and NG8) [24–27]. Recombinant soluble wild-type HA proteins used for ELISA include BR/18 and TS/20.

The optimized coding sequences for wildtype and COBRA proteins in pcDNA3.3 vectors were expressed in soluble proteins following transfection of HEK-293T cells, as described previously [28,29]. Proteins were extracted using HisTrap-Excel columns with the AKTA Pure System (GE Healthcare Bio-Sciences AB, Uppsala, Sweden). Purified proteins were concentrated with phosphate-buffered saline + 0.1% w/v sodium azide (PBSA). Protein concentration was determined using Micro BCA Protein Assay Reagent kits (Pierce Biotechnology, Rockford, IL, USA), and aliquots of each protein were stored at −80°C until used for vaccination.

### Ferrets experiment

Both young adult (9-month-old) and elderly (50–71-month-old) female Fitch ferrets (*Mustela putorius furo*) were sourced from Triple F Farms (Gillett, PA, USA) after de-scenting and spaying (S1A Fig). Ferrets were anesthetized with vaporized isoflurane before blood collection, vaccination, infection, nasal washes, and euthanasia. All animal procedures were performed following the Guide for the Care and Use of Laboratory Animals, Animal Welfare Act and Biosafety in Microbiological and Biomedical Laboratories (AUP: A2020 11–016).

Ferrets (Table 1) were infected intranasally with a mixture of SG/86 and PN/99 influenza viruses ($5 \times 10^5$ per virus/animal) to induce an anti-influenza pre-immune state. Eight weeks after the pre-immune infection, ferrets were vaccinated intramuscularly in the thigh muscle with 15 µg of each protein in a total volume of 250 µL (Fig 1). Infectimune®, (PDS Biotechnology, Princeton, NJ, USA) was mixed in a 1:1 ratio (125 µL sterile 280 mM sucrose with protein: 125 µL Infectimune®). Mock-vaccinated ferrets received 125 µL of sterile 280 mM sucrose with 125 µL of Infectimune®. Four weeks from the prime vaccination, the animals received a booster vaccine of the same mixture. Two weeks after the prime and boost, blood was collected in BD vacutainer SST tubes (BD, Franklin Lakes, NJ, USA). Tubes were incubated for 30 min at RT and then sera was separated by centrifugation at 2500 rpm for 10 min. Purified sera was stored at −20°C.

At 4 weeks after final vaccination, all pre-immune ferrets (both young and elderly) were challenged intranasally with BR/18 H1N1 influenza virus at $1 \times 10^8$ PFU in a volume of 1 mL. Ferrets were monitored daily for weight loss and signs of disease for 14 days post-infection. Nasal wash samples were collected from each ferret on day 3 and 5 post-infection and were stored at −80°C until viral titration.

Animal were humanely euthanized when a cumulative clinical score of three was reached. Clinical scores were determined by nasal discharge/sneezing/diarrhea (0.5; not used for humane endpoint calculation, but used for graphical representation), lethargy (1), dyspnea (2), cyanosis (2), neurological signs (3), moribund state (3), laterally recumbence (3),

**Table 1. Experimental groups.**

| Group | Pre-immune infection | Vaccine | Dose | Adjuvant | No. of Young ferrets | No. of Elderly ferrets |
|---|---|---|---|---|---|---|
| 1 | Sing/86 & Pan/99 | Y2/NG2[a] only | 15 µg/each | None | 4 | 6[b] [44] |
| 2 | Sing/86 & Pan/99 | Y2/NG2[a] | 15 µg/each | Infectimune | 4 | 5[b] [43] |
| 3 | Sing/86 & Pan/99 | Y2/NG5[a] | 15 µg/each | Infectimune | 4 | 3 |
| 4 | Sing/86 & Pan/99 | Y2/NG7[a] | 15 µg/each | Infectimune | 4 | 3 |
| 5 | Sing/86 & Pan/99 | Y2/NG8[a] | 15 µg/each | Infectimune | 4 | 3 |
| 6 | Sing/86 & Pan/99 | Y2/NG5/NG7/NG8[a] | 15 µg/each | Infectimune | 4 | 3 |
| 7 | Sing/86 & Pan/99 | Sucrose | None | None | 4 | None |
| 8 | None | None | None | None | 4 | None |

[a]COBRA antigens.

[b]Data were previously published but were included in this study for additional analysis.

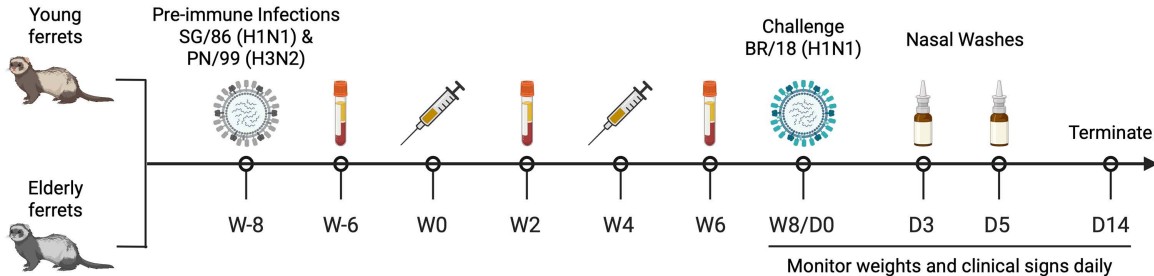

**Fig 1. Experimental design.** Young adult (9 months of age) and elderly (55 months of age on average, range from 50 to 71 months) female Fitch ferrets were made pre-immune with A/Singapore/6/1986 (SG/86) H1N1 and A/Panama/2007/1999 (PN/99) H3N2 influenza viruses, and then eight weeks later these pre-immune elderly ferrets were prime and boost vaccinated with four weeks interval, with the same vaccine formulation. Serum samples were collected after pre-immune infection and each vaccination for analysis. The elderly ferrets were challenged with H1N1 influenza virus at 4 weeks after final boost. Nasal wash samples were collected on day 3 and/or 5 post-infection for viral shedding in nasal cavity. Weights and clinical signs were recorded for all ferrets for a maximum of 14 days post-influenza exposure.

failure to respond to stimuli (3), weight loss of 20–25% (2), and weight loss of >25% (3). The maximum of the two clinical scores recorded for each day was used for analysis.

## Hemagglutination-Inhibition (HAI) assay

The HAI assay was used to determine serum receptor-blocking antibodies elicited in ferrets following vaccination according to the World Health Organization (WHO) laboratory influenza surveillance manual [30]. Briefly, ferret serum samples were treated with receptor-destroying enzyme (RDE) (Denka Seiken, Co., Japan) at 1:3 ratio at 37°C for 18 h and then heat-inactivated at 56°C for 45 min. The RDE-treated serum samples were diluted in a series of two-fold serial dilutions in v-bottom microtiter plates. An equal volume (25 µL) of each H3N2 influenza virus was adjusted to approximately 8 hemagglutination units (HAU)/50 µL in the presence of 20nM Oseltamivir carboxylate was added to each serum serial dilution. For H1N1 influenza viruses, 25 µL of 8 HAU virus was mixed with 25 µL of each serum serial dilution. The plates were covered and incubated at RT for 30 min (for H3N2 assays) or 20 min (for H1N1 assays), followed by the addition of guinea pig red blood cells (GPRBCs) to the H3N2 plates and turkey red blood cells (TRBCs) to the H1N1 plates. The plates were mixed by gentle agitation, covered, and the RBCs were allowed to settle for 1 h (for GPRBCs) or 30 min (for TRBCs) at RT. The HAI titer was determined by the reciprocal dilution of the last well that contained non-agglutinated GPRBCs or TRBCs.

## Influenza viral plaque assay

Influenza virus titers in the nasal washes of infected ferrets were determined by plaque assay as previously described [23]. Briefly, six-well plates were seeded with $1 \times 10^6$ MDCK cells per well one day prior to performing the plaque assay. When cells were 90% confluent in each well, the plates were washed 2x with DMEM supplemented with 1% penicillin-streptomycin (DMEM+P/S) (Thermo Fisher Scientific, Waltham, MA, USA), and infected with 100 µL of each dilution of nasal wash sample. The plates were then incubated at RT for 1 h, shaking every 15 min. After incubation, the supernatant was removed, and cells were washed 2X with fresh DMEM+P/S. Following the second wash, an overlay of 0.8% agarose in 1x MEM (Thermo Fisher Scientific, Waltham, MA, USA), and supplemented with 1 µg/mL of TPCK trypsin (Thermo Fisher Scientific, Waltham, MA, USA) was added into each well. Plates were then incubated at 37°C+5% $CO_2$ for 48–72 h. At the end of incubation, the gel overlays were removed from each well and the cells were fixed with 10% buffered formalin for 15 min and stained with 1% crystal violet (Thermo Fisher Scientific, Waltham, MA, USA) for 15 min at RT. The viral plaques were enumerated as the reciprocal of each dilution. The viral titers were calculated as plaque forming units (PFU)/mL of nasal wash samples.

## Enzyme-linked Immunosorbant Assay (ELISA)

A high-affinity, 96-well flat-bottom enzyme-linked immunosorbent assay (ELISA; Immulon 4HBX) plate was coated with 100 µL of 1 µg/mL of wildtype rHA or rNA in ELISA carbonate buffer (50 mM carbonate buffer [pH 9.5] with 5 mg/mL bovine serum albumin [BSA]), and the plate was incubated overnight at 4°C. The next morning, non-specific epitopes were blocked with 1% BSA in PBS with 0.05% Tween 20 (PBST+BSA) solution for 1 h at RT or overnight at 4°C. Buffer was removed and 3-fold serial dilutions of raw serum were added to the plate with an initial dilution of 1:500. Plates were incubated at 37°C for 90 mins. The plates were washed with PBST, and Goat Anti-Ferret IgG H&L HRP (horseradish peroxidase) (abcam, Waltham, MA, USA) was added at 1:4,000 in PBST+BSA. Plates were incubated at 37°C for 90 min. After washing, 2,29-azino-bis (3-ethylbenzothiazoline-6-sulfonic acid) (ABTS) substrate in McIlvain's buffer (pH 5) was added to each well, and incubated at 37°C for 10 min. The colorimetric reaction was stopped with the addition of 1% SDS in dH2O, and the absorbance was measured at 414 nm using a spectrophotometer (PowerWave XS; BioTek, Winooski, VT, USA).

## Statistical analysis

Data is presented as absolute mean values ± standard deviation (SD). The nonparametric Kruskal-Wallis test, unpaired *t* test or two-way analysis of variance (ANOVA) were used to analyze the statistical differences between groups using GraphPad Prism 10 software (GraphPad, San Diego, CA, USA). A "p" value less than 0.05 was defined as statistically significant (*, P < 0.05; **, P < 0.01; ***, P < 0.001; ****, P < 0.0001).

## Results

### Vaccination with COBRA HA vaccines formulated with Infectimune elicited robust protective antibody responses in pre-immune young and elderly ferrets

Young (9 months old) and elderly (50–71 months old) ferrets were infected with historical influenza viruses to establish pre-existing immunity, mimicking the immune background observed in the human population (Fig 1). All animals seroconverted following infection with SG/86 and PN/99 influenza viruses to establish a pre-immune state (S1 Fig). Ferrets were then vaccinated with COBRA rHA proteins or mock vaccinated. Mock-vaccinated young ferrets had low, but detectable anti-HA antibody titers against the BR/18 HA in serum samples collected after both the prime and boost vaccinations, whereas mock-vaccinated elderly ferrets had low to undetectable anti-HA antibody titers (Fig 2A–B). After the prime vaccination, young ferrets vaccinated with COBRA HA vaccines, with or without Infectimune, had higher IgG titers against BR/18 H1 HA compared to elderly ferrets (Fig 2A). Anti-HA IgG titers against BR/18 HA were maintained in young ferrets following the second vaccination, but antibody titers were increased in elderly ferrets following the booster vaccination (Fig 2B). Anti-HA antibody titers against TS/20 HA followed a similar pattern to those observed for BR/18 HA in both age groups (Fig 2C–D).

Serum samples collected two weeks after boost were tested for receptor-blocking antibodies in a hemagglutination-inhibition assay (Fig 3). Antisera collected from young animals vaccinated with the COBRA HA vaccine alone had, on average, HAI titers ≥80 (6.32 Log$_2$) against all H1N1 viruses isolated from 2009 to 2019, but had, on average, HAI titers of ≥40 (5.32 Log$_2$) against 5 out of 7 H3N2 strains in the panel (Fig 3A). However, there were low to undetectable HAI titers against both H1N1 and H3N2 influenza viruses in elderly ferrets vaccinated with COBRA HA proteins without Infectimune (Fig 3A). In contrast, young ferrets vaccinated with COBRA HA proteins alone had higher HAI titers against the CA/09, BR/18, and GD/19 H1N1 influenza viruses and TS/20 H3N2 influenza virus compared to elderly ferrets (Fig 3A). Young ferrets vaccinated with the H1N1 component Y2 formulated with Infectimune had sera with HAI activity against the panel of recently isolated H1N1 influenza viruses, regardless of which combination of H3N2 COBRA HA proteins used for vaccination (Fig 3B-F). In contrast, elderly ferrets vaccinated with the same vaccines had sera with HAI activity against the panel of recently isolated H1N1 influenza viruses, when it was mixed with either the H3N2 COBRA NG2, NG7, and NG8 HA proteins, but not with H3N2 COBRA NG5 alone or the combination of NG5/NG7/NG8 HA proteins (Fig 3B-F). Young ferrets vaccinated with H3 COBRA NG2 HA vaccine formulated with Infectimune had, on average, HAI titers of ≥40 (5.32 Log$_2$) against 4 out of 7 H3N2 strains (HK/14, SG/16, SA/19, and TS/20) in the panel (Fig 3B). Elderly ferrets also had, on average, HAI titers of ≥40 (5.32 Log$_2$) against 4 out of 7 H3N2 influenza strains (SW/13, HK/14, SA/19, and TS/20) in the panel (Fig 3B). Young ferrets vaccinated with H3 COBRA NG5 HA vaccine formulated with Infectimune had, on average, HAI titers of ≥40 (5.32 Log$_2$) against 2 out of 7 H3N2 strains (HK/19 and TS/20) in the panel (Fig 3C). Elderly ferrets had, on average, HAI titers of ≥40 (5.32 Log$_2$) against 1 out of 7 H3N2 strains (HK/14) in the panel (Fig 3C). Young ferrets vaccinated with H3 COBRA NG7 HA vaccine formulated with Infectimune had average HAI titers of ≥40 (5.32 Log$_2$) against 5 out of 7 H3N2 strains (HK/14, SG/16, HK/19, SA/19, and TS/20) in the panel (Fig 3D). Elderly ferrets had, on average, HAI titers of ≥40 (5.32 Log$_2$) against 3 out of 7 H3N2 strains (KS/17, HK/19, and TS/20) in the panel (Fig 3D). Young ferrets vaccinated with H3 COBRA NG8 HA vaccine formulated with Infectimune had, on average, HAI titers of ≥40 (5.32 Log$_2$) against 2 out of 7 H3N2 strains (HK/19 and TS/20) in the panel (Fig 3E). Elderly ferrets had, on average, HAI titers

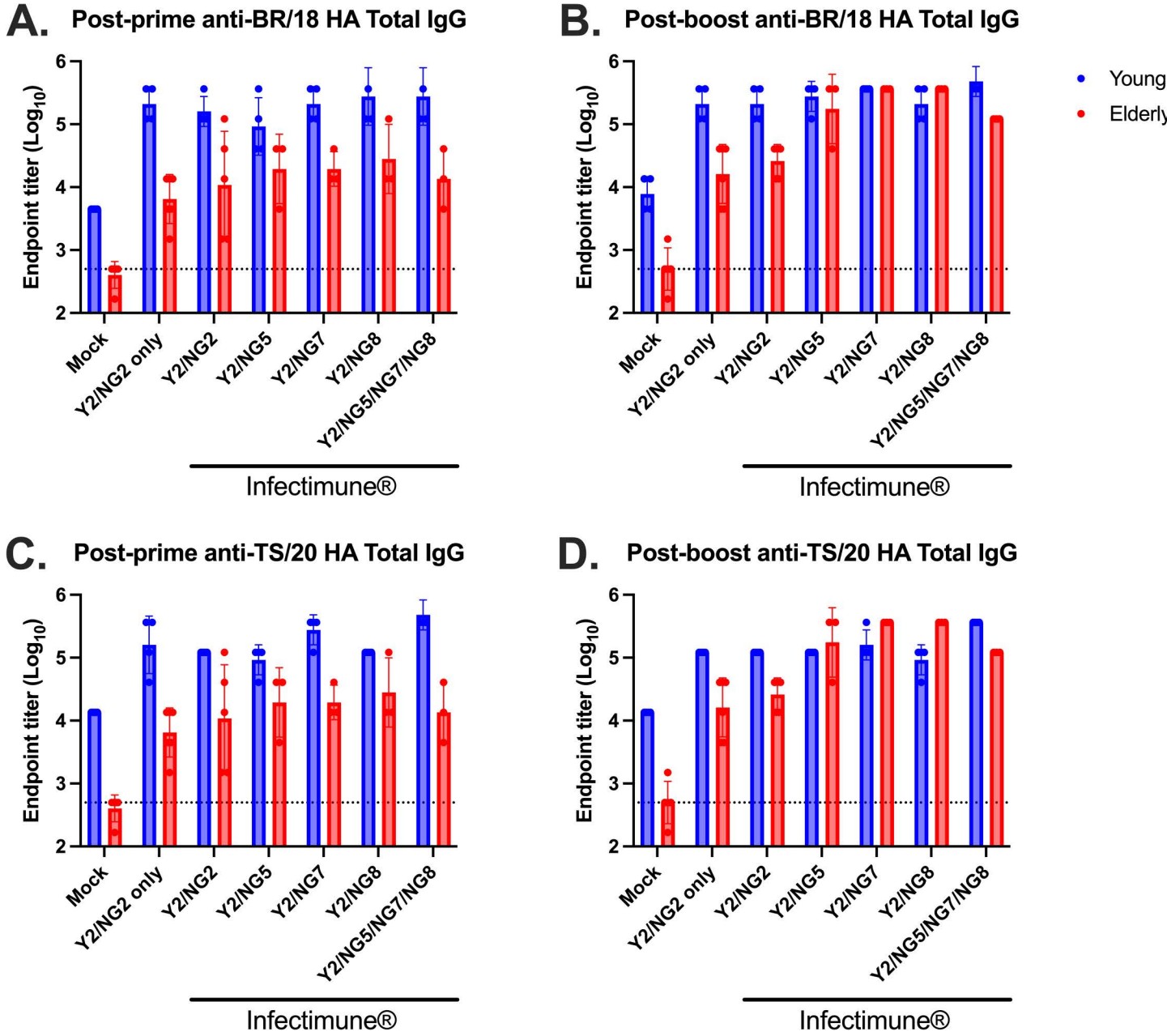

**Fig 2. Total IgG antibody response in both young adult and elderly ferrets after COBRA vaccination.** Sera samples collected 2 weeks after prime (A&C) and boost (B&D) were used in ELISA to determine antibody responses against each strain specific HA after vaccination. The following antigens were used: BR/18 HA (A&B) and TS/20 HA **(C&D)**. Data is presented as average ± standard deviation. The dashed line on the graph indicates limit of detection as 1:500. ELISA data were statistically analyzed using the nonparametric Kruskal-Wallis test, unpaired $t$ test or two-way analysis of variance (ANOVA). A P value of less than 0.05 was defined as statistically significant (*, $P < 0.05$; **, $P < 0.01$; ***, $P < 0.001$; ****, $P < 0.0001$).

of ≥40 (5.32 $Log_2$) against 4 out of 7 H3N2 strains (HK/14, HK/19, SA/19, and TS/20) in the panel (Fig 3E). Pre-immune young and elderly ferrets vaccinated with trivalent H3 COBRA NG5/NG7/NG8 HA vaccine formulated with Infectimune had, on average, HAI titers of ≥40 (5.32 $Log_2$) against same 5 out of 7 H3N2 strains (HK/14, SG/16, HK/19, SA/19, and

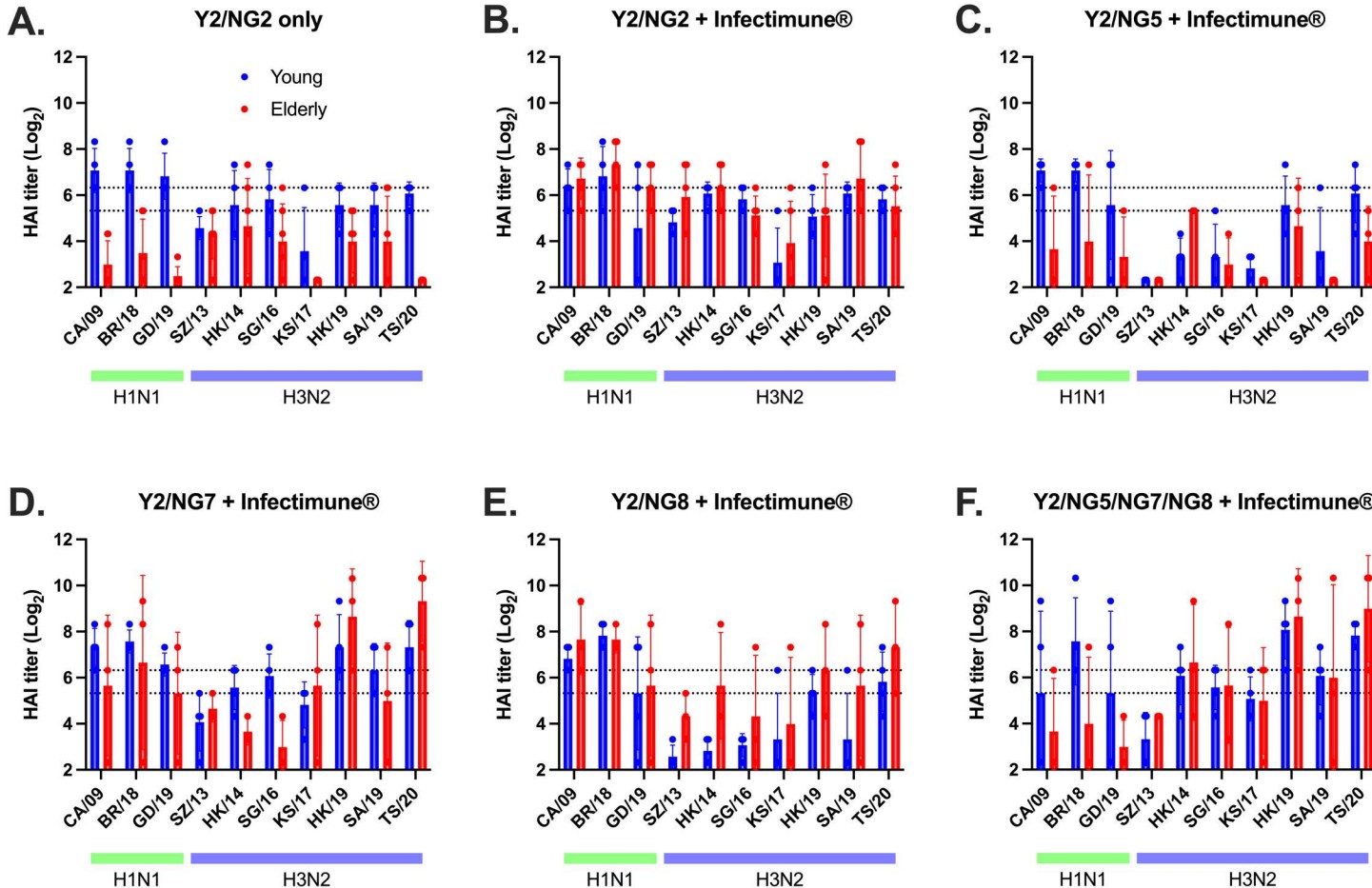

**Fig 3. Hemagglutinin inhibition assays.** Individual ferret serum collected after boost vaccination were used in HAI assay against a panel of historical H1N1 and H3N2 influenza viruses. The title of each figure indicates the vaccine group. Colors indicate young and elderly ferrets. The x-axis indicates influenza virus panel. The y-axis indicates HAI titer in Log$_2$. The lower dashed line indicates 1:40 and the higher dashed line indicates 1:80. Data is presented as average ± standard deviation. HAI titers were statistically analyzed using the nonparametric Kruskal-Wallis test, unpaired *t* test or two-way analysis of variance (ANOVA). A P value of less than 0.05 was defined as statistically significant (*, $P < 0.05$; **, $P < 0.01$; ***, $P < 0.001$; ****, $P < 0.0001$).

TS/20) in the panel (Fig 3F). Overall, these young and elderly ferrets had similar HAI antibodies against the H1N1 and H3N2 influenza viruses, when the vaccine was formulated with Infectimune (Fig 3B-F).

### COBRA HA influenza vaccines formulated with Infectimune mitigated clinical symptoms against H1N1 influenza virus challenge in the young ferrets

To investigate the vaccine-induced protection against influenza virus infection, young ferrets were challenged with BR/18 H1N1 influenza virus four weeks after the booster immunization (Fig 4). Unvaccinated young ferrets (naïve mock) were included as challenge controls for H1N1 influenza virus infection. These animals had ~10% loss of their original body weight by day 8 post-infection. In contrast, pre-immune young ferrets, regardless of vaccination status, had, on average, less than 5% body weight loss following infection (Fig 4A). Naive young ferrets had higher viral titers in nasal wash samples collected on days 3 (~5x10^5 PFU/mL) and 5 (~1x10^4 PFU/mL) post-infection compared with pre-immune young ferrets that received COBRA vaccinations formulated with or without Infectimune (Fig 4B). In comparison, pre-immune

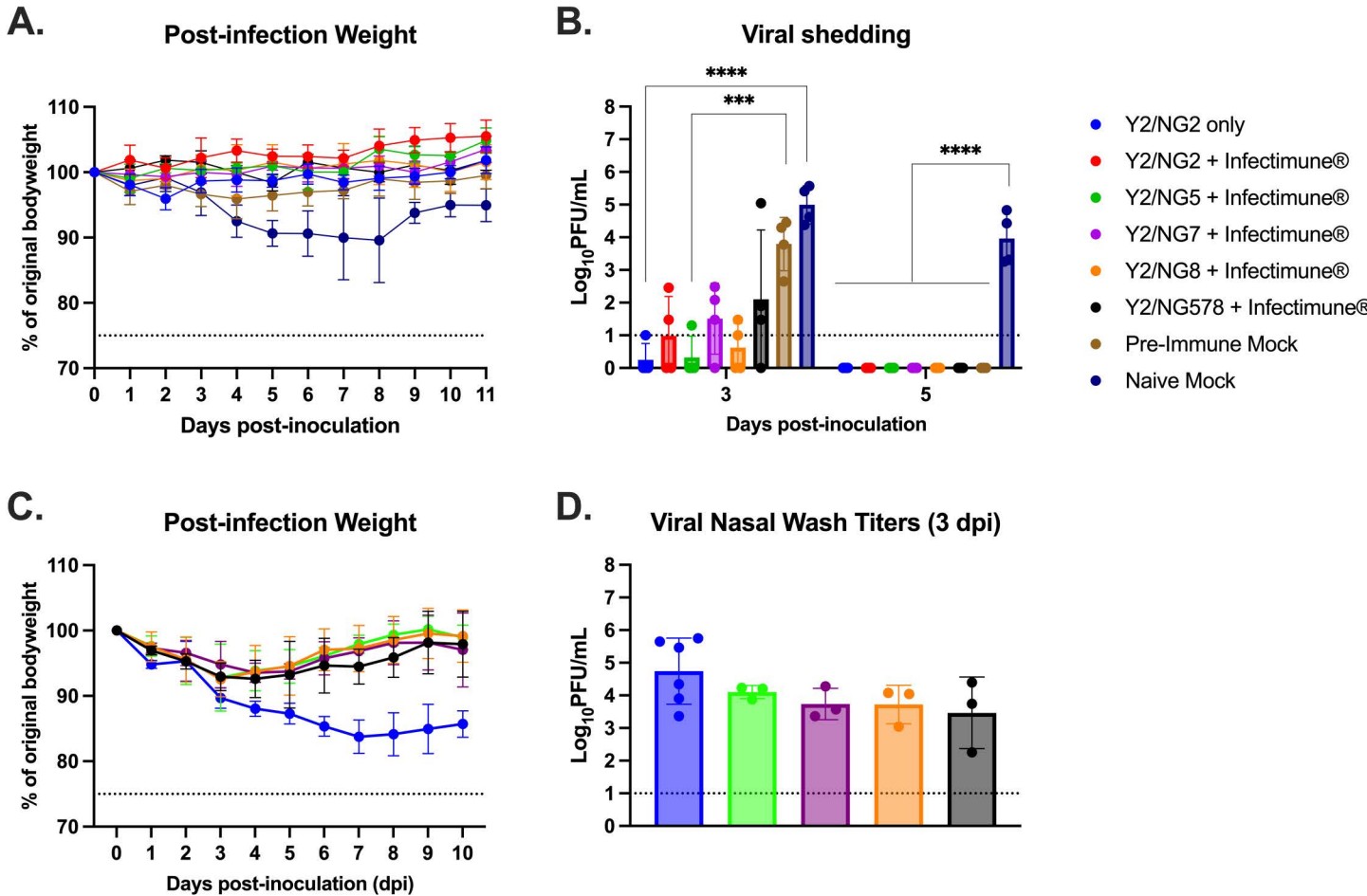

**Fig 4. Young (A & B) and elderly (C & D) ferrets challenged with BR/18 H1N1 influenza virus. (A & C)** The weight loss curves and (B & D) viral shedding in nasal cavity on day 3 and 5 post-infection. Colors indicate experimental groups. Data is given as average ± standard deviation. Statistical analysis was conducted the nonparametric Kruskal-Wallis test, unpaired *t* test or two-way analysis of variance (ANOVA). A P value of less than 0.05 was defined as statistically significant (*, P < 0.05; **, P < 0.01; ***, P < 0.001; ****, P < 0.0001).

mock-vaccinated young ferrets had higher viral titers in nasal wash samples collected on day 3 post-infection, but viral titers were not detectable by day 5 post-infection (Fig 4B). Pre-immune young ferrets, regardless of vaccine formulation, had low to undetectable nasal wash viral titers collected on day 3 post-inoculation, and no animal had detectable virus on day 5 post-infection (Fig 4B). Ferrets vaccinated with COBRA-based vaccines with or without Infectimune had reduced influenza-induced disease symptoms in pre-immune young ferrets. However, the addition of Infectimune further contributed to reduced viral shedding following H1N1 influenza virus challenge.

Following challenge with BR/18, elderly ferrets vaccinated with COBRA HA proteins without Infectimune had ~15% body weight loss by day 7 post-infection (Fig 4C). In contrast, elderly ferrets administered COBRA HA vaccines formulated with Infectimune had a ~5% body weight loss by day 5 post-infection (Fig 4C). Elderly ferrets vaccinated with COBRA HA protein without Infectimune had greater body weight loss compared with elderly ferrets that received COBRA HA vaccines formulated with Infectimune (Fig 4C). All infected elderly ferrets, regardless of vaccine formulation, had detectable virus in collected nasal washes on day 3 post-infection. Viral titers were higher in ferrets vaccinated with COBRA HA alone than in ferrets vaccinated with COBRA HA vaccines formulated with Infectimune (p < 0.05), except in ferrets vaccinated with the

quadrivalent COBRA HA vaccine formulated with Infectimune (Fig 4D). COBRA-based vaccines formulated with Infectimune mitigated influenza virus-induced disease in elderly ferrets, as observed by reduced body weight loss and lower viral titers following challenge with BR/18 H1N1 virus.

## Discussion

A universal influenza vaccine capable of eliciting broadly protective immunity in elderly individuals is urgently needed, as annual vaccination often fails to adequately reduce disease severity in this population. In this study, COBRA-based universal HA vaccines induced broadly protective antibody responses in both young and elderly ferrets and the inclusion of the immune modulator Infectimune, further enhanced the magnitude and protective efficacy of vaccine-induced immunity in aged hosts.

In young hosts, the continuous output of naïve T and B lymphocytes from the thymus and bone marrow helps maintain a diverse immune repertoire within secondary lymphoid tissues such as the spleen [31–33]. However, thymic involution and reduced primary lymphopoiesis in the elderly limit this renewal process [34–36]. Although some lymphocyte development persists in secondary tissues, the composition and functionality of mature lymphocyte pools become profoundly altered, characterized by reduced antibody avidity, a restricted B cell response, skewed CD8+T cell repertoires, and a bias of naïve CD4+T cells toward Th17 rather than Th1 or Th2 differentiation [11,37]. As a result, the aged immune system often requires stronger or repeated stimulation to mount effective adaptive responses against evolving influenza viruses and other respiratory pathogens [11,12]. Consistent with these immunological characteristics, antibody responses to COBRA HA vaccination differed markedly between young and elderly ferrets. In young ferrets, antibody titers rose rapidly after the primary vaccination and remained high throughout the immunization period, reflecting a robust recall response driven by pre-existing immune memory. In contrast, elderly ferrets had weaker primary responses, but showed a pronounced increase in IgG titers only after booster immunization that indicated a delayed activation and expansion of antigen-specific B cells [11,12]. This pattern underscores that age-associated impairments in germinal center reactions and B cell maturation can be partially overcome through booster immunization, which helps restore functional antibody production in elderly hosts [11,12]. Together, these findings highlight fundamental differences in immune responsiveness between young and aged hosts and emphasize the importance of optimized vaccination strategies, such as the use of delivery systems, immune modulators that activate specific immunological pathways, general adjuvants or booster dosing, to enhance protection in elderly populations.

Younger human adults have higher pre- and post-vaccination antibody titers and maintain these titers for longer durations, whereas elderly individuals have both lower baseline and post-vaccination titers, reflecting impaired induction and rapid waning of antibody responses following influenza vaccination [38]. These differences are largely attributed to age-related impairments in germinal center formation, reduced T follicular helper cell support, and diminished generation of long-lived plasma cells and memory B cells [11,12]. The parallel between ferrets and humans underscores the translational relevance of the ferret model for studying immune aging and vaccine efficacy. Collectively, these findings suggest that while vaccination can elicit protective immunity in both age groups, the magnitude and durability of antibody responses are significantly constrained by immunosenescence, highlighting the need for vaccine strategies that can restore or sustain humoral immunity in the elderly.

Although antigenic dominance can be influenced by both the vaccine strain and the assay strain used for evaluation, this study identified that antibody responses were predominantly directed against H1 HA rather than H3 HA in both young and elderly ferrets, indicating a conserved bias in immune recognition that persists across age groups. Previous studies similarly demonstrated that this hierarchy remains stable across diverse vaccine platforms, including mRNA and recombinant protein formulations, and is largely unaffected by adjuvant inclusion suggesting that antigenic imprinting exerts a stronger influence on immune hierarchy than vaccine composition [26,27,39–42]. The finding that both young and elderly ferrets shared comparable antigenic preferences implies that aging does not fundamentally alter HA immunodominance,

but instead attenuates the overall magnitude and functional quality of antibody responses. Given this reduced responsiveness in the elderly, our prior evaluation of a multivalent, high-dose H3 COBRA HA vaccine demonstrated that inclusion of multiple H3 antigens enhanced antibody responses and improved reactivity against H3N2 influenza viruses [43]. Such multivalent strategies may therefore represent a promising approach to augment H3N2-specific immunity and mitigate influenza virus associated morbidity and mortality in aging populations.

Consistent with our previous findings, the NG7 H3 COBRA HA vaccine elicited antibodies with the broadest HAI activity among the vaccine candidates in pre-immune young ferrets [27]. The combination of NG5, NG7, and NG8 HA antigens slightly enhanced antibody responses in both young and elderly ferrets. This multivalent H3 COBRA HA formulation elicited antibody profiles similar to those induced by the single NG7 HA vaccine, indicating that NG7 likely represents a central immunogenic element within the H3 antigenic landscape. The NG5 and NG7 HA sequences differ by seven amino acids distributed across distinct antigenic sites of the H3 HA, which likely account for the major differences in antigenicity observed between these two COBRA HA antigens [27]. By comparison, NG7 and NG8 HA differ by a single amino acid substitution at position 160 within antigenic site B, which introduces a potential glycosylation site [27]. This glycan likely shields a key epitope proximal to the receptor-binding domain in NG8 HA, reducing its accessibility to neutralizing antibodies. The absence of this glycan in NG7 HA may expose this critical epitope, thereby enhance antibody recognition and contribute to the higher HAI titers observed against contemporary H3N2 strains [27].

In summary, COBRA HA vaccination elicited broader and stronger antibody responses in pre-immune young ferrets than in elderly ferrets, though vaccines formulated with Infectimune enhanced responses in both age groups. Young ferrets maintained higher anti-HA IgG and HAI titers against both H1N1 and H3N2 viruses, while elderly ferrets required adjuvanted vaccines to achieve comparable functional antibody levels. Following H1N1 influenza virus challenge, COBRA HA vaccines formulated with Infectimune effectively reduced body weight loss and viral shedding in both young and elderly ferrets. These findings demonstrate that inclusion of immune modulators such as Infectimune in vaccine formulations is critical for improving vaccine immunogenicity and protective efficacy in aged hosts.

## Supporting information

**S1 Fig. Age distribution and pre-immunity.** (A) The age distribution of young and elderly ferrets at the enrollment. (B) HAI antibody response after pre-immune infection. Individual ferret serum collected 2 weeks after pre-immune infections were used in HAI assay against SG/86 H1N1 and PN/99 H3N2 influenza viruses. The y-axis indicates HAI titer in $Log_2$. The lower dashed line indicates 1:40 and the higher dashed line indicates 1:80. Data is presented as average ± standard deviation. Age data and HAI titers were statistically analyzed using one-way analysis of variance (ANOVA) by Prism 10 software. A P value of less than 0.05 was defined as statistically significant (*, $P < 0.05$; **, $P < 0.01$; ***, $P < 0.001$; ****, $P < 0.0001$).
(TIFF)

## Acknowledgments

We thank Spencer Pierce for producing and purifying the HA proteins. We also thank James Allen for designing the next-generation H3 COBRA HA sequences. The authors are grateful to the University of Georgia staff, technicians, and veterinarians for their excellent animal care and support.

## Author contributions

**Conceptualization:** Ted M. Ross.
**Data curation:** Xiaojian Zhang, Matthew H. Thomas, Amanda Lynch, Hua Shi.
**Formal analysis:** Xiaojian Zhang.

**Funding acquisition:** Ted M. Ross.

**Investigation:** Xiaojian Zhang, Matthew H. Thomas, Amanda Lynch, Hua Shi.

**Methodology:** Xiaojian Zhang.

**Project administration:** Xiaojian Zhang.

**Writing – original draft:** Xiaojian Zhang.

**Writing – review & editing:** Xiaojian Zhang, Matthew H. Thomas, Amanda Lynch, Hua Shi, Ted M. Ross.

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
