## [Decision Letter · Decision Letter 0]

23 Nov 2025

Dear Dr. Ross,

We look forward to receiving your revised manuscript.

Kind regards,

Victor C Huber

Academic Editor

PLOS ONE

Journal Requirements:

https://www.clinicalkey.com/#!/content/playContent/1-s2.0-S0264410X25004530?returnurl=https:%2F%2Flinkinghub.elsevier.com%2Fretrieve%2Fpii%2FS0264410X25004530%3Fshowall%3Dtrue&referrer=https:%2F%2Fapi.ithenticate.com%2F

In your revision ensure you cite all your sources (including your own works), and quote or rephrase any duplicated text outside the methods section. Further consideration is dependent on these concerns being addressed.

“This work was supported by the Collaborative Influenza Vaccine Innovations Centers (CIVICs) program of the National Institute of Allergy and Infectious Diseases, National Institutes of Health, Department of Health and Human Services, under contract 75N93019C00052. T.M.R. is also supported in part as a Georgia Eminent Scholar by the Georgia Research Alliance (GRA-001).”

“We thank Spencer Pierce for producing and purifying the HA proteins. We also thank 371 James Allen for designing the next-generation H3 COBRA HA sequences. The authors are 372 grateful to the University of Georgia staff, technicians, and veterinarians for their excellent animal 373 care and support. This work was supported by the Collaborative Influenza Vaccine Innovations 374 Centers (CIVICs) program of the National Institute of Allergy and Infectious Diseases, National 375 Institutes of Health, Department of Health and Human Services, under contract 75N93019C00052. 376 T.M.R. is also supported in part as a Georgia Eminent Scholar by the Georgia Research Alliance 377 (GRA-001)”

“This work was supported by the Collaborative Influenza Vaccine Innovations Centers (CIVICs) program of the National Institute of Allergy and Infectious Diseases, National Institutes of Health, Department of Health and Human Services, under contract 75N93019C00052. T.M.R. is also supported in part as a Georgia Eminent Scholar by the Georgia Research Alliance (GRA-001).”

Additional Editor Comments:

During the revision process, please address the comments related to the presentation of the data, the overall study design as it relates to the limitations associated with the model itself, and the use of statistics in the interpretation of the data presented and conclusions that were made.

Reviewer's Responses to Questions

**Comments to the Author**

1. Is the manuscript technically sound, and do the data support the conclusions?

Reviewer #1: Yes

Reviewer #2: Yes

2. Has the statistical analysis been performed appropriately and rigorously?

Reviewer #1: No

Reviewer #2: Yes

3. Have the authors made all data underlying the findings in their manuscript fully available?

Reviewer #1: Yes

Reviewer #2: Yes

4. Is the manuscript presented in an intelligible fashion and written in standard English?

Reviewer #1: Yes

Reviewer #2: Yes

Reviewer #1: This study examines humoral immunity to P1 and H3 viruses in preimmune ferrets of different ages after immunization with a subunit influenza vaccine based on COBRA hemagglutinin. The chosen animal model appears adequate, and the preimmunization approach is intended to create conditions similar to those of the human population.

The authors should explain in detail the reasons for choosing an 8-week time interval between infection and subsequent immunization.

The number of animals used is not specified anywhere. The total number of animals in the experiment should be indicated. The number of bases in groups should be indicated in the figure legends for Figures 2-4. From the graphs in Figures 2-4, it can be concluded that the number of bases in groups was sometimes 3, sometimes 4. With this number of bases, it is inappropriate to use analysis of variance (ANOVA); the nonparametric Kruskal-Wallis test should be used. In cases where there is a wide range of values (as in Fig. 4, D), doubts arise about any statistically significant differences.

There are no data on the isolation of infectious viruses from the lungs, which would provide valuable information about the course of the infection.

On the one hand, it is clear that the ferret model is expensive, which explains the small number of animals in the groups. On the other hand, it seems that the study was not very well designed from the start. Moreover, the novelty of the finding that the subunit vaccine is ineffective in the elderly population is questionable.

Reviewer #2: The authors present their work with pre-clinical research using animal models to evaluate host immune responses in young versus elderly organisms. This research represents a small but necessary piece of incremental work in the long journey to develop improved flu vaccine regimens, especially for elderly (human) individuals.

I found the paper to be very clear and easy to read. It was well organized, and the study was soundly structured. The results were clear and well explained, and I think the messages in the conclusion were well founded and not out of scope with the study. All in all, I regard this as a tidy and well-done project, it was properly described and reported, and I have no major revisions to recommend w/r/t the textual content of this paper. There were some punctuation errors here and there (e.g., an un-hyphenated "vaccine induced" in line 263), and a couple of other typos or copy-editing quibbles (e.g., "dished line" in line 408; "x-y-axis" in line 416; the abbreviation "TRBC" is never defined), but on the whole I have no major revisions to request. it was a pleasure to review this paper.

The only thing standing in my way of recommending to accept this paper outright (discounting the minor copy edits mentioned above), is one major matter, and I figure that other reviewers are mentioning this as well: in my review copy, Figures 2-4 are illegible, and appear to be the product of taking low-resolution thumbnail images and scaling them up to publication size. I was able to follow along with the Results section, so I believe I was able to make sense of them and understand the authors' descriptions and interpretations, but it was a lot of work and without question these figures need to be revised.

**Do you want your identity to be public for this peer review?** For information about this choice, including consent withdrawal, please see our Privacy Policy

Reviewer #1: No

Reviewer #2: No

---

## [Author Response · Author response to Decision Letter 1]

7 Dec 2025

December 6, 2025

To: Editor – PLoS One

Manuscript title: Assessment of Immunogenicity and Protection Induced by Adjuvanted COBRA HA Vaccines in Young and Elderly Ferrets

Ref: Submission ID: PONE-D-25-58584

Dear Editor:

We have updated the manuscript (Ref: PONE-D-25-58584), entitled “Assessment of Immunogenicity and Protection Induced by Adjuvanted COBRA HA Vaccines in Young and Elderly Ferrets” by Zhang and Ross submitted to PLoS One based upon the referee comments. The changes requested were clearly indicated in the track-changes version of the revised manuscript. We have prepared a list of answers to the Reviewers’ comments below.

Reviewer #1

Comment #1: This study examines humoral immunity to P1 and H3 viruses in preimmune ferrets of different ages after immunization with a subunit influenza vaccine based on COBRA hemagglutinin. The chosen animal model appears adequate, and the preimmunization approach is intended to create conditions similar to those of the human population.

Response: Thank you for the comments.

Comment #2: The authors should explain in detail the reasons for choosing an 8-week time interval between infection and subsequent immunization.

Response: From past experiences and publications, an interval of less than 6 weeks does not allow enough time for the infection induced immune responses to decline and can interfere with vaccination induced response. We have standardized our pre-immune infection model to an 8-week interval following pre-immune infections prior to vaccination.

Comment #3: The number of animals used is not specified anywhere. The total number of animals in the experiment should be indicated. The number of bases in groups should be indicated in the figure legends for Figures 2-4. From the graphs in Figures 2-4, it can be concluded that the number of bases in groups was sometimes 3, sometimes 4. With this number of bases, it is inappropriate to use analysis of variance (ANOVA); the nonparametric Kruskal-Wallis test should be used. In cases where there is a wide range of values (as in Fig. 4, D), doubts arise about any statistically significant differences.

Response: We have added a table of group information in the text: Line 475-477

We have updated statistical analysis with nonparametric Kruskal-Wallis test and revised figures and figure legends to indicate the updated statistical analyses.

Comment #4: There are no data on the isolation of infectious viruses from the lungs, which would provide valuable information about the course of the infection.

Response: We thank the referee for the comments, however, only nasal washes were performed to determine viral titers following challenge.

Comment #5: On the one hand, it is clear that the ferret model is expensive, which explains the small number of animals in the groups. On the other hand, it seems that the study was not very well designed from the start. Moreover, the novelty of the finding that the subunit vaccine is ineffective in the elderly population is questionable.

Response: Thank you for the comments.

Reviewer #2: The authors present their work with pre-clinical research using animal models to evaluate host immune responses in young versus elderly organisms. This research represents a small but necessary piece of incremental work in the long journey to develop improved flu vaccine regimens, especially for elderly (human) individuals.

Comment #1: I found the paper to be very clear and easy to read. It was well organized, and the study was soundly structured. The results were clear and well explained, and I think the messages in the conclusion were well founded and not out of scope with the study. All in all, I regard this as a tidy and well-done project, it was properly described and reported, and I have no major revisions to recommend w/r/t the textual content of this paper.

Response: Thank you for the positive comments.

Comment #2: There were some punctuation errors here and there (e.g., an un-hyphenated "vaccine induced" in line 263), and a couple of other typos or copy-editing quibbles (e.g., "dished line" in line 408; "x-y-axis" in line 416; the abbreviation "TRBC" is never defined), but on the whole I have no major revisions to request. it was a pleasure to review this paper.

Response: We have revised the entire manuscript for punctuation errors and updated as needed. TRBC abbreviation stands for turkey red blood cells and this reference is now listed on page 8, line 197-198.

Comment #3: The only thing standing in my way of recommending to accept this paper outright (discounting the minor copy edits mentioned above), is one major matter, and I figure that other reviewers are mentioning this as well: in my review copy, Figures 2-4 are illegible, and appear to be the product of taking low-resolution thumbnail images and scaling them up to publication size. I was able to follow along with the Results section, so I believe I was able to make sense of them and understand the authors' descriptions and interpretations, but it was a lot of work and without question these figures need to be revised.

Response: The quality of the figures has been adjusted to meet the journal’s guidelines.

Sincerely,

Ted M. Ross

Global Director of Vaccine Development

Florida Research & Innovation Center

Cleveland Clinic

Email: rosst7@ccf.org

---

## [Editor Report · Decision Letter 1]

10 Dec 2025

Assessment of immunogenicity and protection induced by COBRA HA vaccines formulated with Infectimune in young and elderly ferrets

PONE-D-25-58584R1

Dear Dr. Ross,

We’re pleased to inform you that your manuscript has been judged scientifically suitable for publication and will be formally accepted for publication once it meets all outstanding technical requirements.

Kind regards,

Victor C Huber

Academic Editor

PLOS One
---

## [Editor Report · Acceptance letter]

PONE-D-25-58584R1

PLOS One

Dear Dr. Ross,

I'm pleased to inform you that your manuscript has been deemed suitable for publication in PLOS One. Congratulations! Your manuscript is now being handed over to our production team.

Kind regards,

on behalf of

Dr. Victor C Huber

Academic Editor

PLOS One